# The Effect of Genotyping on the Number of Pharmacotherapeutic Gene–Drug Interventions in Chronic Kidney Disease Patients

**DOI:** 10.3390/pharmacy11020069

**Published:** 2023-04-04

**Authors:** Catharina H. M. Kerskes, Carien J. M. E. van den Eijnde, Albert-Jan L. H. J. Aarnoudse, René J. E. Grouls, Birgit A. L. M. Deiman, Maarten J. Deenen

**Affiliations:** 1Department of Clinical Pharmacy, Catharina Hospital, 5623 EJ Eindhoven, The Netherlands; 2Faculty of Pharmacy, University Utrecht, 3508 TB Utrecht, The Netherlands; 3Department of Nephrology and Dialysis, Catharina Hospital, 5623 EJ Eindhoven, The Netherlands; 4Department of Clinical Chemistry, Catharina Hospital, 5623 EJ Eindhoven, The Netherlands; 5Department of Clinical Pharmacy and Toxicology, Leiden University Medical Centre, 2333 ZA Leiden, The Netherlands

**Keywords:** pharmacogenetics, chronic kidney disease, medication evaluation, polypharmacy

## Abstract

Patients with chronic kidney disease (CKD) stage 3–5 are polypharmacy patients. Many of these drugs are metabolized by cytochrome P450 (*CYP450*) and *CYP450.* Genetic polymorphism is well known to result in altered drug metabolism capacity. This study determined the added value of pharmacogenetic testing to the routine medication evaluation in polypharmacy patients with CKD. In adult outpatient polypharmacy patients with CKD3-5 disease, a pharmacogenetic profile was determined. Then, automated medication surveillance for gene–drug interactions was performed based on the pharmacogenetic profile and the patients’ current prescriptions. Of all identified gene–drug interactions, the hospital pharmacist and the treating nephrologist together assessed clinical relevance and necessity of a pharmacotherapeutic intervention. The primary endpoint of the study was the total number of applied pharmacotherapeutic interventions based on a relevant gene–drug interaction. A total of 61 patients were enrolled in the study. Medication surveillance resulted in a total of 66 gene–drug interactions, of which 26 (39%) were considered clinically relevant. This resulted in 26 applied pharmacotherapeutic interventions in 20 patients. Systematic pharmacogenetic testing enables pharmacotherapeutic interventions based on relevant gene–drug interactions. This study showed that pharmacogenetic testing adds to routine medication evaluation and could lead to optimized pharmacotherapy in CKD patients.

## 1. Introduction

The prevention of adverse drug events (ADEs) is of especial importance in patients with chronic kidney disease (CKD), since they may result in further loss of kidney function and patient harm. Because CKD patients often have multiple comorbidities, including hypertension, diabetes mellitus, anaemia, coronary artery disease, etc., the treatment of these comorbidities is associated with a high pill burden [1]. Moreover, high pill burden is associated with higher risk of ADEs that may lead to the addition of even more drugs for treatment of these ADEs [2,3].

Most drugs used in CKD patients are largely excreted by the kidney, and impaired renal clearance of drugs mostly necessitates dose adjustments or the substitution of contraindicated drugs [4]. If no adequate dose adjustments or substitutions are made, this may lead to adverse drug events. In order to optimize pharmacotherapeutic treatment, CKD patients visiting our outpatient nephrology clinic that are not yet on dialysis routinely receive medication evaluations once every 6 months. These medication evaluations are performed by a hospital pharmacist. This hospital pharmacist is part of a multidisciplinary team of nephrologists, nurse practitioners, dieticians and social health workers providing care to this group of patients.

Prior to excretion by the kidney, most of applied drugs undergo extensive metabolism by the cytochrome P450 (CYP450) enzyme family. It is well known that the metabolic activity of CYP-enzymes varies between individuals due to, e.g., genetic variations in *CYP450* genes, clinical, drug and food interactions, and basic patient characteristics, such as age and gender. These genetic variations in CYP-enzymes can lead to a different metabolic activity and a different response to specific drugs metabolized by this CYP-enzyme. A lower metabolic activity of CYP-enzymes can lead to side effects of drugs metabolized by this enzyme, or less activity in case of a prodrug that needs activation by this enzyme. A higher metabolic activity of CYP-enzymes can lead to less or even no effect of drugs metabolized by this specific enzyme. For prodrugs that need activation or drugs metabolized to a more potent metabolite, a higher metabolic activity can lead to side effects.

Whereas the application of (pre-therapeutic) pharmacogenetic testing became part of routine care in oncology, cardiology and psychiatry in most countries [5,6], in nephrology, this remains limited to CKD patients eventually receiving kidney transplantation and being treated with immunosuppressant drugs such as tacrolimus [7,8,9]. One paper addressed the potential role for pharmacogenomics testing to optimize antihypertensive regimens in patients with CKD [10]. To the best of our knowledge, no studies were conducted that determined the added value of systematic pharmacogenetic testing in a general CKD patient population. We hypothesized that *CYP450* genetic polymorphism in patients with CKD affects drug response differently than would solely be expected from the genetic profile alone. Therefore, in this study, we aimed to gain more insight into the potential value of systematic pharmacogenetic testing during the routine medication evaluation in CKD patients. The potential value was defined as the number of pharmacotherapeutic interventions based on a relevant gene–drug interaction.

## 2. Materials and Methods

### 2.1. Study Design

This was a prospective single-centre interventional study. The patient population consisted of adult polypharmacy patients with CKD3-5 disease, who were not on dialysis, who visited the multidisciplinary CKD outpatient clinic of the Catharina Hospital, Eindhoven, The Netherlands. Inclusion criteria were CKD3-5 disease, no requirement of dialysis, age 18 years or older, and the use of 5 of more oral or parenteral drugs. There were no specific exclusion criteria. All patients participating in the study provided written informed consent. The study was approved by the Medical Research Ethics Committees United (MEC-U), The Netherlands, with registration number W19.066.

### 2.2. Study Procedure

In each patient enrolled in the study, we determined a pharmacogenetic profile consisting of *CYP1A2*, *CYP2B6*, *CYP2C9*, *CYP2C19*, *CYP2D6*, *CYP3A4*, *CYP3A5* and *VKORC1*. This was carried out in already-obtained, leftover EDTA whole blood for routine laboratory measurements. Genotype data were translated into phenotypes according to the Dutch Pharmacogenetics Working Group (DPWG) guidelines [11]. In these guidelines, the criteria for poor metabolizer (PM), intermediate metabolizer (IM), normal metabolizer (NM), rapid metabolizer (RM), ultrarapid metabolizer (UM) and increased inducability (II) were derived from the metabolic activity of both alleles according to specific variants. For example, *CYP2C19 *17/*17* lead to an augmented metabolic activity and a UM phenotype. After translation according to the DPWG guidelines, the phenotypes of all CYP450 enzymes and VKORC1 were recorded into the electronic health record (EHR; HiX, Chipsoft, Amsterdam, The Netherlands) for each individual patient, allowing automated medication surveillance based on gene–drug interactions. For each individual patient, all currently used drugs were electronically prescribed in the EHR as part of routine care. The combination of an individuals’ drug prescriptions and genotype data allowed the EHR to automatically identify all gene–drug interactions according to the DPWG guidelines [10]. For all patients, all gene–drug interactions were herewith retrieved. Subsequently, the clinical relevance of all identified gene–drug interactions were discussed by the hospital pharmacist with the treating nephrologist. If the observed gene–drug interaction was considered clinically relevant according to the discretion of both the hospital pharmacist and the treating nephrologist, this gene–drug interaction and subsequent change in drug regimen was discussed with the patient. The change in drug regimen was divided into three groups, i.e., discontinuation of the drug, switch of the drug or dose adjustment of the drug. Figure 1 provides an overview of the study procedure.

### 2.3. Genotype Analysis

Genomic DNA was isolated from 200 µL EDTA whole blood using the MP24 Total Nucleic Isolation kit (Roche Diagnostics B.V. Almere, The Netherlands) on the MagNAPure 24 system (Roche Diagnostics) using the Fast hgDNA 200 3.1 protocol. Of the DNA extracts, 4 µL was mixed with 4 µL Taqman^®^ Genotyping Master Mix (Thermofisher, Waltham, MA, USA). Of this suspension, 5 µL was placed on a TaqMan^®^ Open Array^®^ PGx Express Panel (Thermofisher), detecting *CYP1A2*(**1C*, **1D*, **1E*, **1F*, **1K*), *CYP2B6*(**5*, **6*, **16*, **22*), *CYP2C19*(**2*, **3*, **4*, **5*, **6*, **7*, **8*, **9*, **10*, **11*, **17*), *CYP2C9*(**2*, **3*, **4*, **5*, **6*), *CYP2D6*(**2*, **3*, **4A*, **6*, **7*, **8*, **9*, **10*, **12*, **14*, **17*, **29*, **41*), *CYP3A4*(**1B*, **2*, **3*, **12*, **17*, **22*), *CYP3A5*(**2*, **3*, **3B*, **6*, **7*, **8*, **9*) and *VKORC1-1639G* > *A*, using the Accufill system (Thermofisher) and tested on the QuantStudio™ 12K Flex instrument (Thermofisher) according to the Pharmacogenomics Experiments Application guide (MAN0009612, Thermofisher). Data were analysed using the QuantStudio™ 12K Flex software (Thermofisher). To determine the *CYP2D6* copy number variation, 1 µL of the DNA extracts was mixed with 9 µL reaction mix including TaqPath™ ProAmp™ Master Mix, TaqMan^®^ Copy Number Assay (exon 9), TaqMan^®^ Copy Number Reference Assay, and nuclease-free water, according to Pharmacogenomics Experiments Application guide (MAN0009612, Thermofisher), on a MicroAmp^®^ Fast 96-well Reaction Plate. Each sample was tested four times. Amplification was performed on the QuantStudio™ 12K Flex instrument (Thermofisher), according to the Pharmacogenomics Experiments Application guide (MAN0009612, Thermofisher). Data analysis was performed using the CopyCaller v2.1 software (Thermofisher).

## 3. Results

### 3.1. Study Population and Baseline Characteristics

In the period between November 2018 and March 2019, a total of 61 patients were included who were visiting the consulting hospital pharmacist from the multidisciplinary team. The mean age of the study population was 73 years. The median eGFR in the study population was 18 mL/min. Patients used a median (range) number of 12 (5–20) drugs. All baseline characteristics of the study population are shown in Table 1. The drugs that were used most frequently with a minimum frequency of 10% in all patients are shown in Table 2.

### 3.2. Pharmacogenetic Testing

Pharmacogenetic testing within the eight pharmacogenes in the 61 patients resulted in a total of 185 altered phenotypes, and are provided in Table 3.

Phenotypes other than the normal phenotype were observed in 27 patients (44%) for CYP1A2, 27 patients (44%) for CYP2B6, 36 patients (59%) for CYP2C19, 14 patients (23%) for CYP2C9, 32 patients (52%) for CYP2D6, 6 patients (10%) for CYP3A4, 6 patients (10%) for CYP3A5 and 37 patients (60%) for VKORC1.

### 3.3. Gene–Drug Interactions and Pharmacotherapeutic Interventions

Based on these 185 altered phenotypes in the 61 patients and their currently used drugs, automated medication surveillance identified a total of 66 potential gene–drug interactions. Of these potential gene–drug interactions, 26 gene–drug interactions in 20 patients (33%) were considered as clinically relevant by the hospital pharmacist and the treating nephrologist. This corresponds with a number needed to genotype of 3. Table 4 shows the clinical implications of the gene–drug interactions for the individual patient. A gene–drug interaction was only considered clinically relevant if the patient either experienced a side effect of this drug or if the efficacy of that specific drug was considered less effective, and of which the association was possibly, potentially or definitively related. As an example, participant number 12 had hypoglycemic periods on a normal dose of gliclazide possibly due to a slower metabolism of gliclazide being a CYP2C9 intermediate metabolizer [12]. Therefore, this drug was switched to another drug. Participant number 37—being a CYP2D6 poor metabolizer—experienced no effect of tramadol. Notably, tramadol requires CYP2D6-mediated activation to a much more potent active metabolite and, thereby, provides the explanation for the lack of effectiveness of this drug [13]. Participant number 49—being a CYP2D6 intermediate metabolizer—experienced bradycardia and complained of dizziness during the day. Therefore, the dose of metoprolol was lowered [14].

The gene–drug interactions that resulted in a pharmacotherapeutic intervention were derived from various gene–drug combinations. The most common gene–drug combinations were with CYP2D6-beta blocker interactions (31%), CYP2C19-proton pump inhibitor interactions (15%) and CYP2C19-clopidogrel interactions (8%). All of the identified gene–drug interactions that were considered clinically relevant resulted in pharmacotherapeutic interventions and change in the drug regimens. These interventions consisted of a dose adjustment (10 times, 39%), a drug discontinuation (7 times, 27%) and drug switch (9 times, 34%).

## 4. Discussion

In this study, we assessed the added value of systematic pharmacogenetic testing to the complete medication evaluation in CKD patients. The results of this study showed that other than the general medication evaluation, pharmacogenetic testing led to additional pharmacotherapeutic interventions in one third of the patients, corresponding with a number needed to genotype of 3. This is considered a relatively low number and in addition, may even implicate a favourable cost–benefit profile of systematic pharmacogenetic testing in these patients [15].

Previous studies in general patient populations determined the frequency and type of potential gene–drug interactions according to electronic health records [16,17,18]. These potential gene–drug interactions may or may not lead to side effects or no effect of the corresponding drug. In our study, we also determined the frequency and type of potential gene–drug interactions, and this study is the first to do so in a CKD population. As well as determining the frequency, we were additionally able to determine the clinical relevance of these possible gene–drug interventions by discussing the identified gene–drug interaction with the nephrologist and the patient. This pharmacotherapeutic consult was conducted by the hospital pharmacist and was based on the reported side effects or no effect by the individual patient during the frequent visits to the CKD outpatient clinic.

As well as general patient characteristics such as age and gender affecting CYP450 activity, CKD is one of the chronic diseases that additionally may impact CYP450 metabolism [19]. Namely, patients with CKD retain uremic solutions, consisting of protein-bound molecules (e.g., indoxyl sulfate and hippuric acid), free water-soluble low molecular weight molecules such as reactive carbonyl compounds and middle molecular weight molecules including parathyroid hormone and cytokines. The retention of uremic solutions leads to accumulation of uremic toxins. Of note, multiple studies showed that uremic toxins decrease CYP450 activity [19,20,21]. Thereby, this reduced CYP450 activity specifically in CKD patients directly impacts CYP450-mediated drug metabolism. Considering this already reduced CYP450 activity in CKD patients, in addition to the already-reduced renal function, genetic alterations in CYP450 encoding genes may be of especial importance in this patient population. This may potentially similarly act as so-called phenoconversion. This means that the individual’s predicted genotype drug metabolism capacity differs from its true capacity. Phenoconversion may result from, e.g., concomitant CYP450-inhibiting drugs, increasing age, inflammation, but also disease, such as, in this case, CKD [22]. Notwithstanding, this subject remains largely uninvestigated in the CKD patient population.

The earlier in the process of kidney decline the systematic pharmacogenetic testing in these patients is performed, the more useful it may become to prevent future adverse drug events. As kidney function declines, the side effects of a drug using a specific metabolism on which a pharmacogenetic variation is observed might become more prominent.

For future research, it would be interesting to investigate a few model compounds that predominantly depend on renal excretion and have a sole CYP metabolization route in which therapeutic drug monitoring could be performed in patients with impaired renal function to objectify this hypothesis.

In this study, we used the internationally well-recognized DPWG guidelines to, first, translate the pharmacogenetic testing result into the corresponding phenotypes and, second, to retrieve potential gene–drug interactions based on the current drug prescriptions for the individual patient. We used this guideline because it is the designated guideline used in most electronic health records in The Netherlands. Additionally, other guidelines such as the CPIC guidelines exist. The use of another guideline may potentially lead to another result of potential gene–drug interactions because of the differences between these guidelines in the number and described clinical relevance of gene–drug interactions [23]. However, because of the final step in our study in which we assessed the clinical relevance of the potential gene–drug interactions by using patient reports of side effects or no effect, we believe that potential differences between guidelines were largely overcome.

One of the limitations of our study was that we included patients of our nephrology outpatient clinic, regardless of the period they were already visiting this clinic. In addition, only the potential gene–drug interactions of the current medication patients were using were determined. Therefore, we could not consider the historically executed pharmacotherapeutic interventions. Stopping medication or dose adjustments of medication already conducted in the past because of adverse drug events may have also been caused by a relevant gene–drug intervention. This may have led to an underreporting of the number of pharmacotherapeutic interventions in our study.

Another limitation of our study was the rather short duration of follow-up that did not allow us to report the patient outcomes after changing the drug regimen. For future research, this would be an interesting addition to the knowledge gained so far. In addition, patient satisfaction with the changes in drug regimen after pharmacogenetic testing would also be of interest for further investigation.

## 5. Conclusions

Systematic pharmacogenetic testing in CKD patients leads to additional pharmacotherapeutic interventions based on relevant gene–drug interactions and has a low number needed to genotype. This study showed that pharmacogenetic testing in CKD patients can be of added value to the routine medication evaluation, and showed good potential to further optimize pharmacotherapy in this polypharmacy patient population. It is important to extend future research to the clinical benefit of these pharmacotherapeutic interventions including patient reported outcomes.

## Figures and Tables

**Figure 1 pharmacy-11-00069-f001:**
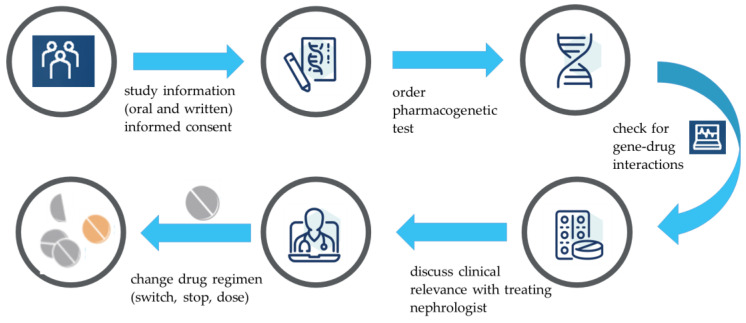
Overview of the study procedure.

**Table 1 pharmacy-11-00069-t001:** Baseline characteristics of the study population.

Characteristic	*n* (%)
Gender	
male	38 (62%)
female	23 (38%)
Age, mean (range)	73 (47–86)
Race (caucasian)	61 (100%)
Smoking	
yes	5 (8%)
no	56 (92%)
eGFR (CKD-EPI mL/min), median (range)	18 (9–30)
Number of drugs, median (range)	12 (5–20)
Comorbidities	
hypertension	41 (67%)
heartfailure	41 (67%)
atrial fibrillation	10 (16%)
diabetes	22 (36%)

Abbreviations: eGFR = estimated Glomerular Filtration Rate; CKD-EPI = Chronic Kidney Disease Epidemiology Collaboration.

**Table 2 pharmacy-11-00069-t002:** Most frequently used drugs.

Drug	*n*	(%)
Colecalciferol	57	(93%)
Pantoprazole	34	(56%)
Metoprolol	32	(52%)
Acetylsalicylic acid	26	(43%)
Allopurinol	23	(38%)
Alfacalcidol	20	(33%)
Darbepoetin alfa	20	(33%)
Furosemide	19	(31%)
Simvastatin	19	(31%)
Amlodipine	15	(25%)
Folic acid	14	(23%)
Macrogol	14	(23%)
Acenocoumarol	12	(20%)
Acetaminophen	12	(20%)
Rosuvastatin	12	(20%)
Sodiumbicarbonate	11	(18%)
Atorvastatin	10	(16%)
Ferrofumarate	10	(16%)
Hydrochlorothiazide	9	(15%)
Lercanidipin	9	(15%)
Lisinopril	9	(15%)
Spironolacton	9	(15%)
Prednisolone	8	(13%)
Bisoprolol	7	(11%)
Bumetanide	7	(11%)
Clopidogrel	7	(11%)
Doxazosine	7	(11%)
Ezetimibe	7	(11%)
Linagliptin	7	(11%)
Perindopril	7	(11%)

**Table 3 pharmacy-11-00069-t003:** Pharmacogenetic variants translated into phenotype according to the Dutch Pharmacogenetic Working Group guidelines.

*n* = 61	PM	IM	NM	RM	UM	II
*CYP1A2*	0	2	25	n.a.	0	34 *^1^
*CYP2B6*	3	24	34	n.a.	0	n.a.
*CYP2C19*	3	19	25	12	2	n.a.
*CYP2C9*	1	13	47	n.a.	0	n.a.
*CYP2D6*	9	23	29	n.a.	0	n.a.
*CYP3A4*	0	6	55	n.a.	0	n.a.
*CYP3A5*	55	4	2	n.a.	n.a.	n.a.
*VKORC1*	9	28	24	n.a.	n.a.	n.a.

Abbreviations: PM = poor metabolizer; IM = intermediate metabolizer; NM = normal metabolizer; RM = rapid metabolizer; UM = ultrarapid metabolizer; II = Increased inducability (*^1^ relevant in smoking population); n.a. = not applicable.

**Table 4 pharmacy-11-00069-t004:** Overview of all interventions based on observed clinically relevant gene–drug interactions.

Subject	Intervention	CYP	Phenotype	Drug with Gene–Drug Interaction	Clinical Implication *
6	switch	CYP2C19	IM	clopidogrel	less effect
12	switch	CYP2C9	IM	gliclazide	side effect
	dosage	VKORC1/CYP2C9	PM/IM	acenocoumarol	side effect
17	switch	VKORC1	IM	fenprocoumon	side effect
18	dosage	CYP2D6	IM	metoprolol	side effect
	discontinuation	CYP2C19	IM	pantoprazole	side effect
20	switch	CYP2C19	UM	pantoprazole	less effect
	discontinuation	CYP2C19	UM	amitriptylin	less effect
22	switch	CYP3A4	IM	simvastatin	side effect
23	discontinuation	CYP2C19	IM	clopidogrel	less effect
	discontinuation	CYP3A5	IM	amiodarone	less effect
24	dosage	CYP2D6	IM	bisoprolol	side effect
25	switch	CYP3A4	IM	simvastatin	side effect
	dosage	CYP2D6	IM	metoprolol	side effect
27	dosage	CYP2D6	IM	risperidone	side effect
33	dosage	CYP2D6	IM	metoprolol	side effect
	switch	CYP2C9	PM	glimepiride	side effect
36	dosage	CYP2D6	IM	bisoprolol	side effect
37	discontinuation	CYP2D6	PM	tramadol	less effect
47	dosage	CYP2C19	IM	pantoprazole	side effect
49	dosage	CYP2D6	IM	metoprolol	side effect
51	dosage	CYP2D6	IM	metoprolol	side effect
55	switch	CYP3A4	IM	simvastatin	side effect
59	discontinuation	CYP2D6	PM	tramadol	less effect
60	switch	CYP2C19	UM	pantoprazole	less effect
61	dosage	CYP2D6	IM	metoprolol	side effect

Abbreviations: CYP = Cytochrome P450, IM = intermediate metabolizer, UM = ultrarapid metabolizer, PM = poor metabolizer. * A gene–drug interaction was only considered clinically relevant if the patient either experienced a side effect of this drug or if the efficacy of that specific drug was considered less effective, and of which the association was possibly, potentially or definitively related.

## Data Availability

The data presented in this study are available on request from the corresponding author. The data are not publicly available due to privacy and ethical reasons.

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
