# Peer review of "The Effect of Genotyping on the Number of Pharmacotherapeutic Gene–Drug Interventions in Chronic Kidney Disease Patients"

_pharmacy, 2023, doi:10.3390/pharmacy11020069_

Round 1

Reviewer 1 Report (New Reviewer)

The authors describe the use of PGx in the chronic kidney disease population.  Overall, this is a small study.  While drug-genotype interactions were identified, it is not clear to me genotyping improved patient outcomes other than providing added support where there was decreased efficacy or adverse drug reaction.  Recommend adding implication to discussion.

Major concerns:

Note that there is at lease another published study on chronic kidney disease and PGx genotyping.  KIDNEY360 3: 307–316, 2022. doi: https://doi.org/10.34067/KID.0005362021.  Please review and consider related study in this manuscript.

What are the patient demographics regarding race/ethnicity in your study population?  Please add to your patient demographics.  This can impact PGx testing as this reviewer noted that several African ancestral variants were not tested for CYP2C9.  This may be OK if the patients were of white/European ancestry.  

Minor comments:

Consider using the term variant/variants instead of polymorphism and mutation(s).  

page 8; line 203 - add space between proton and pump

Author Response

Please see the attachment. We kindly thank you for your suggestions to improve our manuscript. 

Reviewer 2 Report (New Reviewer)

1.      Can the authors discuss a bit how the patients were sub divided into different metabolizers group?

2.      Page 6 line 194: …more much more… please correct.

3.      In table 4, for ‘switch’ intervention, what does the ;drug’ column represent? The one which is switched from or switch to? Can the authors include a column for the one the ;drug’ column does not represent? What about discontinuation? Does that affect the patient’s overall health condition?

4.      Page 8 line 203: insert space between ‘protonpump’

5.      Can the author elaborate the conclusion and includes some implication and application of this in the context of future studies?

Author Response

Please see the attachment. We kindly thank you for your suggestions to improve our manuscript. 

This manuscript is a resubmission of an earlier submission. The following is a list of the peer review reports and author responses from that submission.

Round 1

Reviewer 1 Report

Main point

1. The explanation for tables is insufficient. In particular, table 4 is the most important, but the results and explanation do not match. Table 1 does not reflect the characteristics of study population sufficiently. There is no foot note for abbreviations.

2. The authors provided the limitations. I think this is the main point for this paper. The authors should present the specific status and clinical course. If possible, the follow-up period of the patients should be increased.

Minor point

1. The authors should use the abbreviation of chronic kidney disease after the full words.

2. The authors should present the number of approved MEC-U.

Reviewer 2 Report

The research aimed to clarify the impact of genetic variants on the effectiveness and safety of the drug. Therefore, it is of interest and significance. However, I would recommend the following comments that should be addressed.

1. Introduction: It would be helpful for the readers to follow if adding some background information about cytochrome P450 enzyme and genetic variations.

2. Methods: The criteria for PM, IM, NM, RM, UM, II should be addressed.

3. Results and discussions:

(1) It seems that there is not enough data and evidence to convince the readers that the study achieved the goal stated in the introduction “Therefore, in this study we aimed to determine the added value of systematic pharmacogenetic testing during the routine medication evaluation in CKD patients, defined as the number of pharmacotherapeutic interventions based on a relevant gene-drug interaction.”

(2) Line 161, ------26 gene-drug interactions in 21 patients, but 20 subjects were included in Table 4. Please clarify it, also as calculations from lines 173 to 178.

(3) Please clarify the relationship between the genetic variants and drugs such as CYP2D6 and beta blocker, CYP2C19 and proton-pump inhibitor, CYP2C19 and clopidogrel, based on the data or literature.

(4) More information about pharmacotherapeutic intervention would be helpful for the audiences to follow, mainly why the intervention was performed in patients with no effect in clinical implication.

4. To be consistent, please capitalize overview in Table 4.

5. Please carefully check and correct the text. For example,

(1) the word he doesnt seem to fit the context in line 245;

(2) Consider adding a comma after the introductory phrase For future research.
